# Influence of Coarse Grain Content on the Mechanical Properties of Red Sandstone Soil

Junhua Chen [1,2], Yanjiang Zhang [1], Yanxin Yang [1,*], Bai Yang [1,*], Bocheng Huang [1] and Xinping Ji [1]

[1] School of Architecture and Transportation Engineering, Guilin University of Electronic Technology, Guilin 541004, China
[2] Traffic and Transportation Engineering Postdoctoral Mobile Station, School of Civil Engineering, Central South University, Changsha 410075, China
* Correspondence: yanxinyangswjtu@foxmail.com (Y.Y.); ayangbai@163.com (B.Y.)

**Abstract:** Coarse-grained red sandstone soil is often used as embankment filling material but is prone to being broken by extrusion, which lowers the stability of the roadbed. This paper aimed to clarify the influence of the variation in coarse-grain content on the mechanical properties of coarse-grained red sandstone soil. Soil with a grain size greater than 5 mm is regarded as coarse-grained soil, and coarse-grained red sandstone soils with different contents of coarse grains were prepared as cylindrical specimens with a diameter of 300 mm and a height of 600 mm. Under three different confining pressures, a large-scale triaxial apparatus was used to carry out triaxial shear tests. The results showed that as the content of coarse grains of red sandstone (denoted as $p$) increased, the deviation stress of static failure increased, showing a hyperbolic relationship. The internal friction angle also increased hyperbolically, while the cohesion reached a peak value and then decreased, and the maximum value of 133.8 kPa was reached at $p = 30\%$. As the content of coarse grains increased, the maximum dilatancy increased. The maximum amount of shrinkage reached a peak value and then decreased, and the maximum value was reached when $p = 30\%$. A coarse grain content $p$ equal to 30% was the optimum value when coarse-grained red sandstone soil was used as a filling material.

**Keywords:** soil mechanics; triaxial shear test; deviation stress of static failure; cohesion; internal friction angle; elastic modulus





## 1. Introduction

Coarse-grained soil is a granular accumulation that has not been completely weathered, composed of soil and stone, and it is commonly used as a roadbed filling for important construction projects [1–3]. Changes in the ingredients of the parent rock coarse-grained soil will lead to variation in the physical structure and mechanical properties after compaction and molding of the coarse-grained soil, which will affect the stability of the subgrade.

Red sandstone soil is widely distributed in Hunan, Jiangxi, Sichuan, Guizhou, and Yunnan provinces in China. Considering the economic factors and environmental protection factors, red sandstone soil is often directly used as a roadbed filler [4,5]. The strength of red sandstone soils is generally lower. When used as a subgrade filler, under compaction and molding, coarse-grained red sandstone soil is easy to break. According to previous research [6,7], the larger the grain size of red sandstone soil, the more likely it is to be crushed, caused by extrusion. Zhang et al. [8] found that due to the weak occlusal effect between fine grains, the fine grain easily deforms under external loads, and the bearing capacity decreases significantly under bias loads. The grain size and content of soil have an important influence on the mechanical properties of coarse-grained red sandstone soil. Therefore, it is necessary to systematically analyze the influence of the coarse-grain content on the mechanical properties of soil.

Bagherzadeh–Khalkhali and Mirghasemi [9] investigated the effect of grain size on the shear strength of coarse-grained soils based on direct shear tests and discrete element

numerical simulations. Zhang et al. [10] studied the dynamic properties of coarse-grained soils under cyclic triaxial loading with different coarse-grain contents of permafrost. Liu et al. [11] investigated the effect of grain size on the small-strain shear modulus of coarse-grained soils. Wen et al. [7] used the discrete unit method to simulate biaxial compression tests and found that the shear strength of coarse-grained soils with the same specimen size and different maximum grain sizes increased with the increase in maximum grain size. However, the mechanical properties test of coarse-grained soil has been mainly carried out using large instruments, such as via the large triaxial shear test, large direct shear test and large simple shear test. The large-scale triaxial shear test is currently the most widely used test in the study of engineering mechanical properties of soil [4,12–15]. Wang et al. [16] investigated the swelling properties of coarse-grained soils, using Lade's dilatancy equation to predict the swelling rate of coarse-grained soils and verifying it with the results of large-scale triaxial tests. Babenko [17] and Voznesensky et al. [14] studied the relationship between stress and the volumetric strain of coarse-grained soils using the true triaxial test. It was found that the deviatoric component plays an important role in affecting the volumetric compression of coarse-grained soil and that, in areas of stress concentration, coarse-grained fragmentation promotes further rearrangement of the particles and suppresses the increase in volumetric strain. Zhang et al. [18] conducted consolidation drained (CD) and consolidation undrained (CU) shear tests by using a large-scale triaxial test instrument to study the rheological characteristics of sericite quartz schist coarse-grained soil. Seo et al. [19] used a large-scale direct shear test and a large-scale triaxial test to analyze the shear strength of coarse-grained soil. Through these tests, the cohesion, internal friction angle, elastic modulus, stress–strain curve, dilatancy, shear shrinkage and other mechanical properties can be obtained.

Analyzing the above results, it can be seen that the stress–strain relationship, the shear strength and the volume changes in coarse-grained soils have been studied by some scholars through experiments. However, the current research on coarse-grained red sandstone soils is not deep enough, especially concerning the changes in mechanical properties of red sandstone soils caused by the coarse-grain content, which requires further research. Therefore, according to the research conducted by Guo [20], 5 mm was used as the grain size limit between coarse grains and fine grains. In this paper, large consolidated drained triaxial tests were carried out on specimens with different coarse-grain contents to obtain various mechanical parameters of red sandstone soils with different coarse-grain contents, including deviatoric stress, cohesion, angle of internal friction, modulus of elasticity, swelling and shrinkage for static damage. At the same time, this paper analyzed the effect of coarse-grain content on the shear performance of red sandstone as a common road base filler and determined the optimum ratio of coarse and fine grains in red sandstone soils, which provides some performance reference for red sandstone coarse-grained soils used as fillers.

## 2. Triaxial Shear Test

### 2.1. Test Instrument

In this paper, the large-scale static and dynamic triaxial apparatus TAJ-2000 was used to carry out the triaxial shear test on coarse-grained red sandstone soil, and the apparatus was produced by Tianshui Hongshan. The instrument is shown in Figure 1. The test system was digitally controlled, including the axial compression system, confining pressure system, and water and vapor systems. The working principle diagram of the instrument is shown in Figure 2. The maximum force that could be applied axially on the sample was 2000 kN; both the maximum confining pressure and the maximum pore water pressure were 5 MPa, and the maximum volumetric strain was ±0.1%. The loading pattern included force-control and strain-control modes. An axial force was applied to the bottom surface of the cylinder, and confining pressure was applied to the surface of the cylinder. The large-scale triaxial test loading diagram is shown in Figure 3.

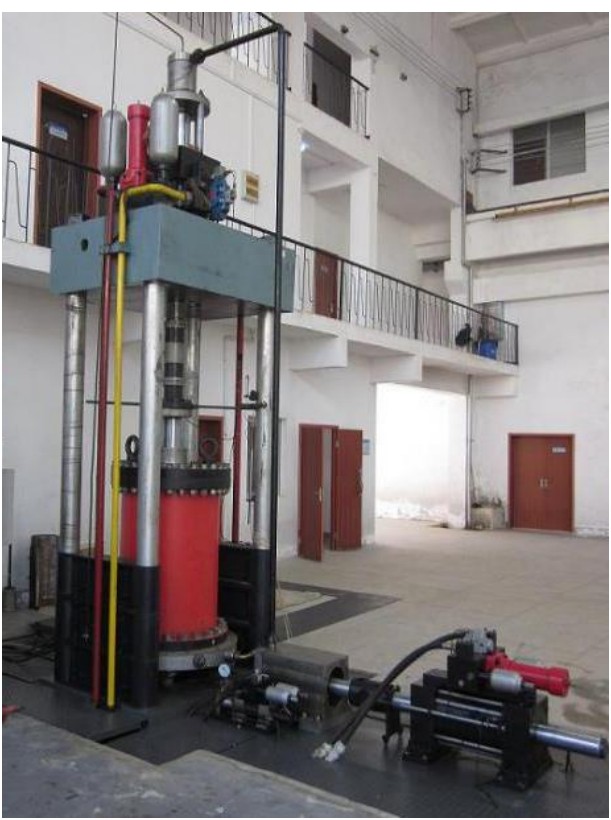

**Figure 1.** The appearance of the test instrument.

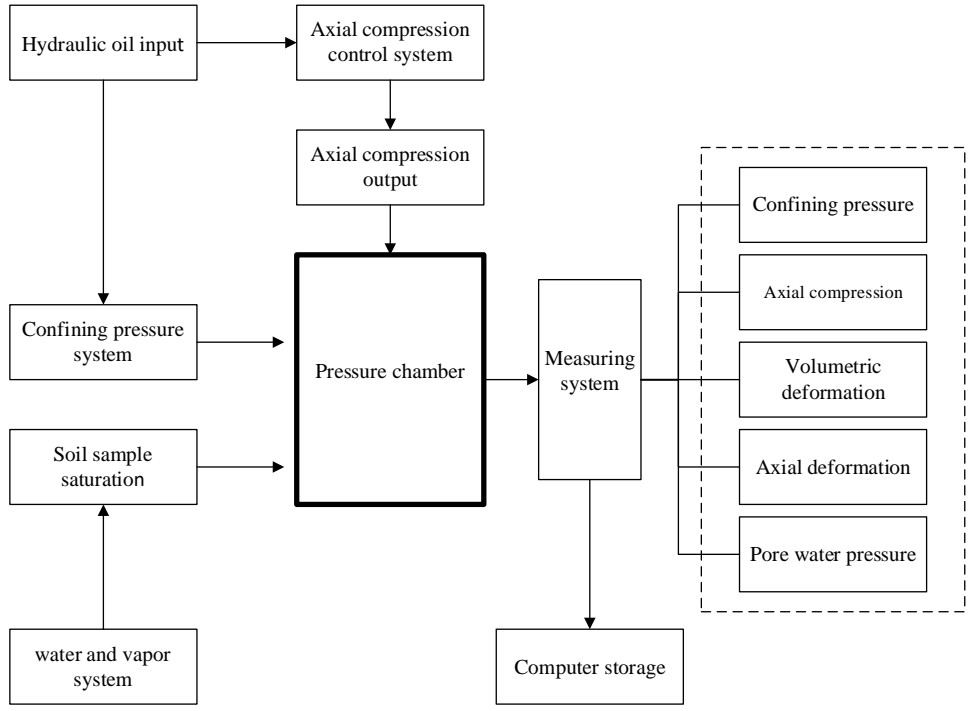

**Figure 2.** The working principle diagram of the test instrument.

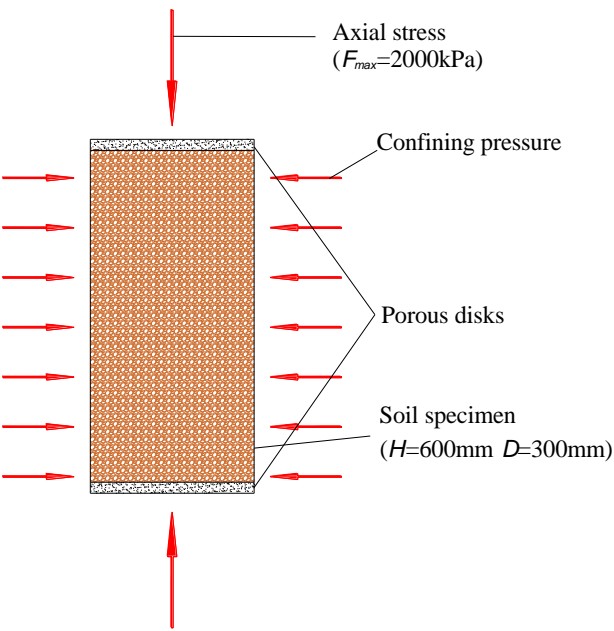

**Figure 3.** Large-scale triaxial test loading diagram.

### 2.2. Sample Preparation

The sample soil was coarse-grained and obtained from Changsha, Hunan Province, and its parent rock was fully weathered red sandstone. The main parameters of the physical properties of the red sandstone coarse-grained soils are shown in Table 1. As the soil strength was low, it was disintegrative and easily crushed by extrusion. Taking a 5 mm grain size as the boundary between coarse grains and fine grains, the mass percentages of coarse grains with a grain size of more than 5 mm in coarse-grained red sandstone soil were 10%, 30%, 50%, 70% and 90%, denoted as *p*. The coarse-grained soil was prepared into a cylinder with a diameter of 300 mm and a height of 600 mm, and there were 15 samples in total.

**Table 1.** The main parameters of the physical properties of the red sandstone coarse-grained soils.

| Natural Moisture Content (%) | Natural Density (g/cm$^3$) | Specific Gravity $G_s$ | Plastic Limit $W_P$ (%) | Liquid Limit $W_L$ (%) | Plasticity Index $I_P$ | Void Ratio $e$ |
|---|---|---|---|---|---|---|
| 7.9 | 2.05 | 2.58 | 21.5 | 33.3 | 11.8. | 0.39 |

According to the Chinese standard Code for Soil Test of Railway Engineering (TB10102-2010) [21], the compaction tests were carried out using a DJ30-5 heavy-duty compaction machine, using the rejection method for oversized particles. The compaction work was undertaken at 2701.4 kJ/m$^3$ in three passes. The maximum dry density and optimum moisture content were obtained from the compaction curves. The grading curves of the samples are shown in Figure 4. The maximum dry density and optimum moisture content of each sample are shown in Table 2. Due to the large size of the sample, it was difficult to place the sample after compaction; therefore, the sample was directly prepared on the base of the pressure chamber by using the open semi-circular forming barrel. The soil sample was prepared as shown in Figure 4 and Table 2. The sample was compacted into five layers, filled layer by layer with the test material; each layer was compacted with a steel rod; then, the test material was brought to the calculated height with a percussion hammer, and the soil compaction ratio was 0.9. The compacted sample is shown in Figure 5.

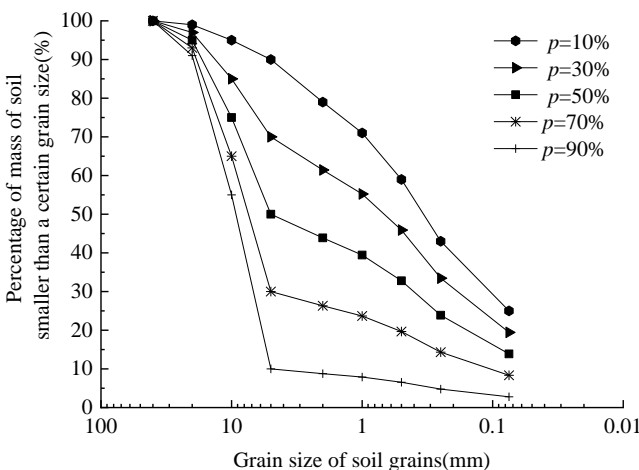

**Figure 4.** Grading curves of samples with different mass percentages of coarse grains.

**Table 2.** Maximum dry density and optimum moisture content.

| The Mass Percentage of Coarse Grains $p$ (%) | Maximum Dry Density (g/cm$^3$) | Optimum Moisture Content (%) |
|---|---|---|
| 10 | 1.894 | 9.32 |
| 30 | 2.185 | 8.63 |
| 50 | 1.956 | 5.34 |
| 70 | 1.908 | 4.01 |
| 90 | 1.867 | 3.45 |

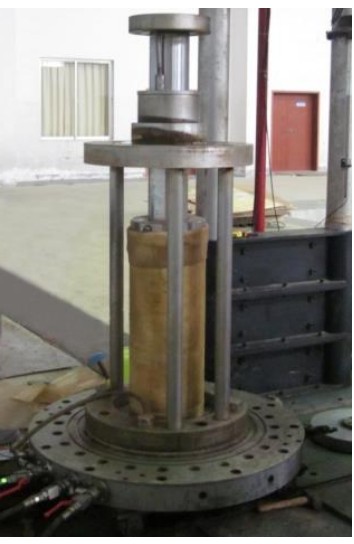

**Figure 5.** The compacted soil sample.

### 2.3. Test Loading Scheme

The confining pressure was defined as *S*. The confining pressure in this test *S* equaled 200 kPa, 350 kPa and 600 kPa. Triaxial shear tests were carried out on samples with different coarse-grain sizes under various confining pressures. In the test, firstly, isotropic consolidation tests were carried out by applying the axial pressure and confining pressure. Then, the confining pressure remained unchanged, and the axial load was applied to compress the specimen using the strain-control mode. According to the testing standard in [21], the axial strain should be controlled at 0.1% to 0.5% per minute. In this paper, the loading rate was 0.3% per minute (i.e., velocity was 1.8 mm per minute), and the sample

was in a drainage state during the loading process. The samples before and after loading are shown in Figures 6 and 7, respectively.

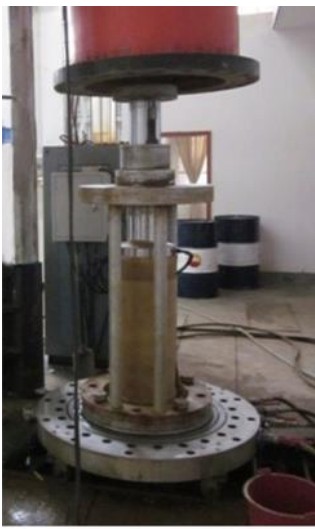

**Figure 6.** The sample before loading.

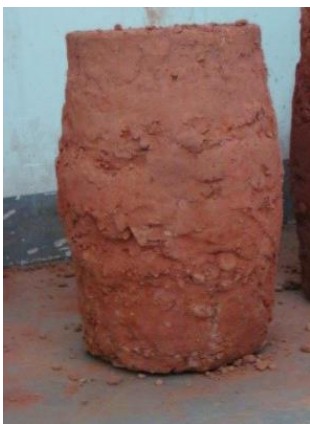

**Figure 7.** The sample after loading.

## 3. Test Results and Analysis

### 3.1. Stress–Strain Relationship Analysis of Coarse-Grained Soil

The axial stress was defined as $\sigma_1$ and the axial strain was defined as $\varepsilon_1$. In Figure 8, the typical strain-softening stress–strain curves for the confining pressure of 200 kPa are shown. Taking the curve corresponding to $p$ = 10% as an example, the curve can be divided into three segments. For description, the end points and intermediate interval points of the curve were represented by the letters *O*, *A*, *B*, and *C*, where *B* was the point where the maximum axial stress occurs. In the initial stage of loading, as is shown in the *O-A* segment of the curve, the strain was small, and the relationship was linear. As the strain $\varepsilon_1$ continued to increase, in the *A-B* segment of the curve, the localized soil grains gradually produced a dislocation, slip, and overturn phenomenon. The cracks in the soil grains penetrated through the whole grain breakage phenomenon [22], thus forming a shear band. The stress growth rate began to slow down, and the $\sigma_1 \sim \varepsilon_1$ relationship gradually changed from linear to nonlinear. As is shown in point *B*, when the shear band in the soil reached the critical state of failure, the stress reached the maximum. Subsequently, after the original structure of the shear band was destroyed, the soil grain breakage occurred significantly, and the grains began to rearrange. As is shown in the *B-C* segment curve, with the increase in strain, the stress gradually decreased. When the rearrangement of soil grains tended to be

completed, the soil recovered its ability to resist deformation, and the soil stress gradually tended to be stable.

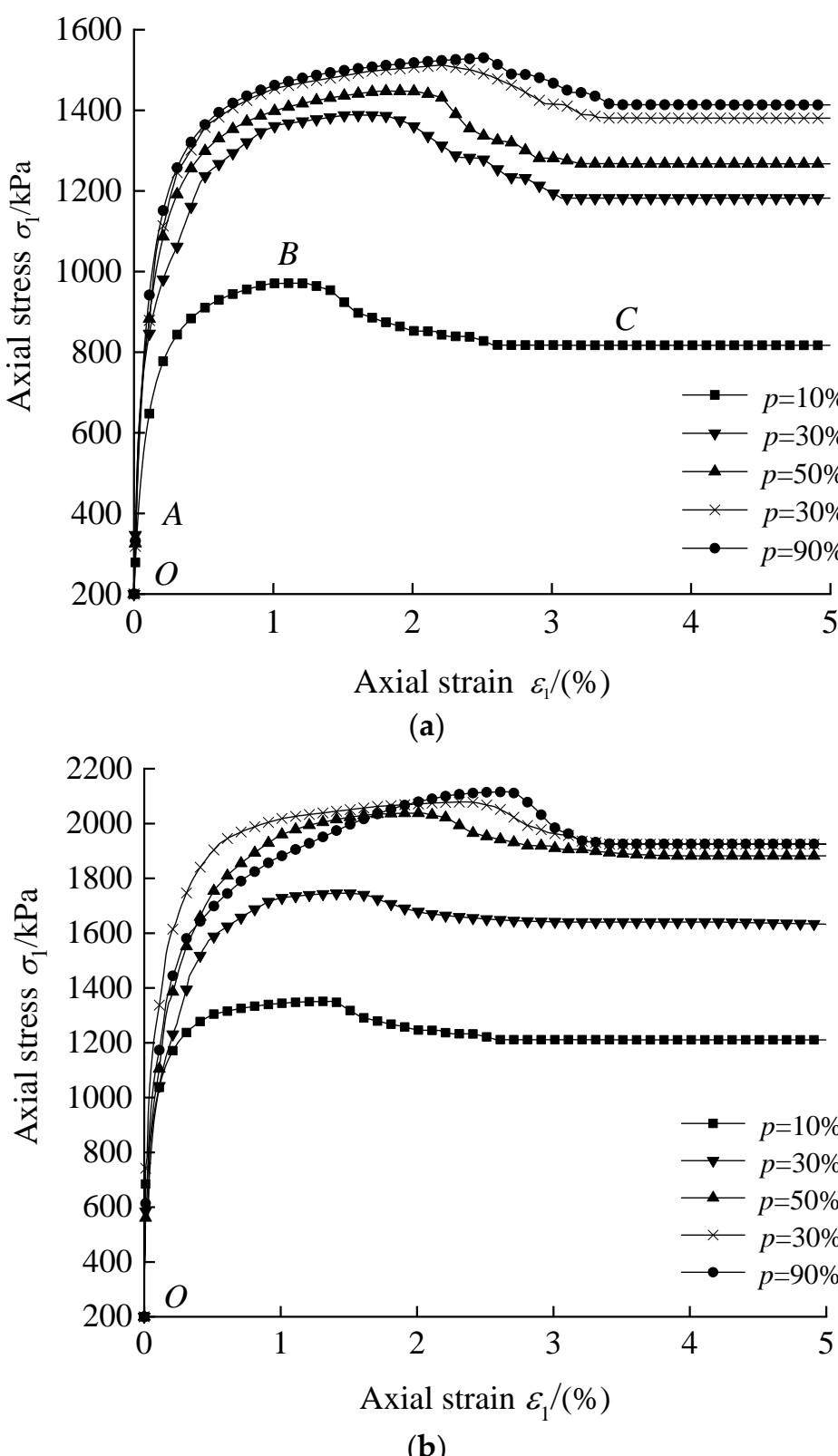

**Figure 8.** *Cont.*

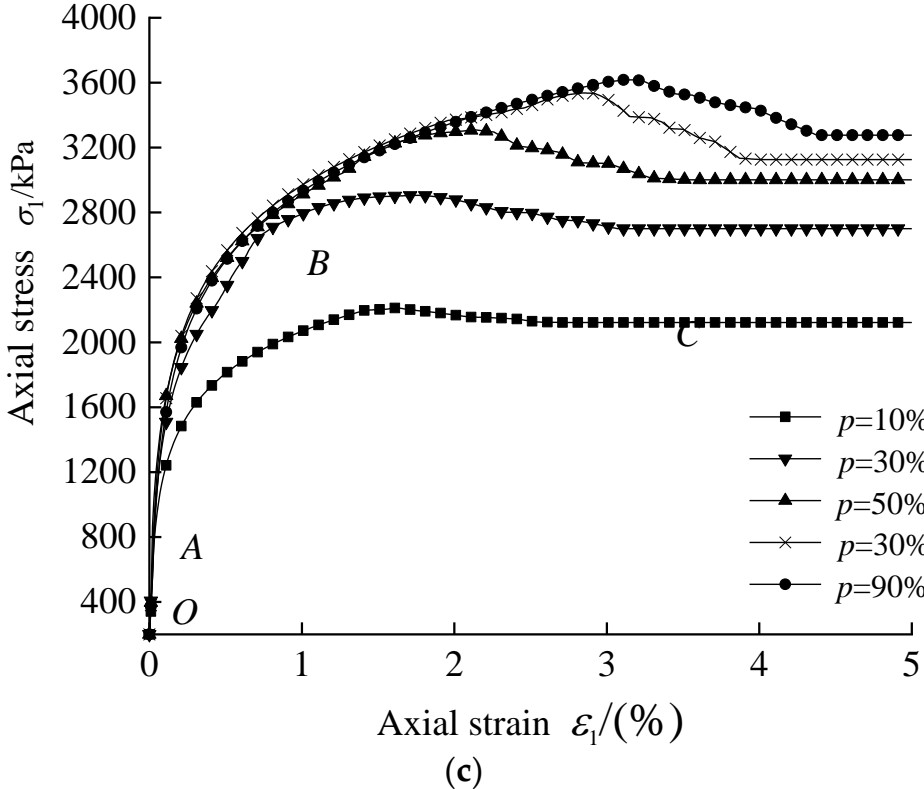

**Figure 8.** The curves of the relationship between axial stress and axial strain. (**a**) S = 200 kPa, (**b**) S = 350 kPa, (**c**) S = 600 kPa.

### 3.2. The Effect of Coarse Grain Content on the Deviation Stress of Static Failure

The deviation stress of static failure was used to measure the shear strength. As the test in this paper was an isotropic consolidation shear test, the deviation stress equaled the axial stress minus the confining pressure. If the deviation stress of static failure is $Y$, then $Y$ satisfies Equation (1):

$$Y = \sigma_{1\max} - S \tag{1}$$

where $\sigma_{1\max}$ is the maximum value of axial stress, $S$ is the confining pressure, and $Y$ is the deviation stress of static failure.

The strength of the soil is influenced by both fine grains and coarse grains in the soil. With the increase in the coarse grain content, the influence of coarse grains on the maximum value of the axial stress gradually becomes greater. The relationship $Y\sim p$ between the deviation stress of static failure of the sample and the mass percentage of coarse grains under different confining pressures is shown in Figure 9. The deviation stress of static failure first increased and then gradually stabilized with the increase in the mass percentage of coarse grains. As $p$ increased from 10% to 90%, when confining pressure $S$ was equal to 200 kPa, the deviation stress of static failure $Y$ increased from 770.9 kPa to 1330.6 kPa by a percentage of 72.6%. When the confining pressure $S$ was equal to 350 kPa, the deviation stress of static failure $Y$ increased by a percentage of 76.5% from 1000.5 kPa to 1765.5 kPa. When the confining pressure $S$ was equal to 600 kPa, the deviation stress of static failure $Y$ increased by a percentage of 87.2% from 1612.3 kPa to 3018.2 kPa.

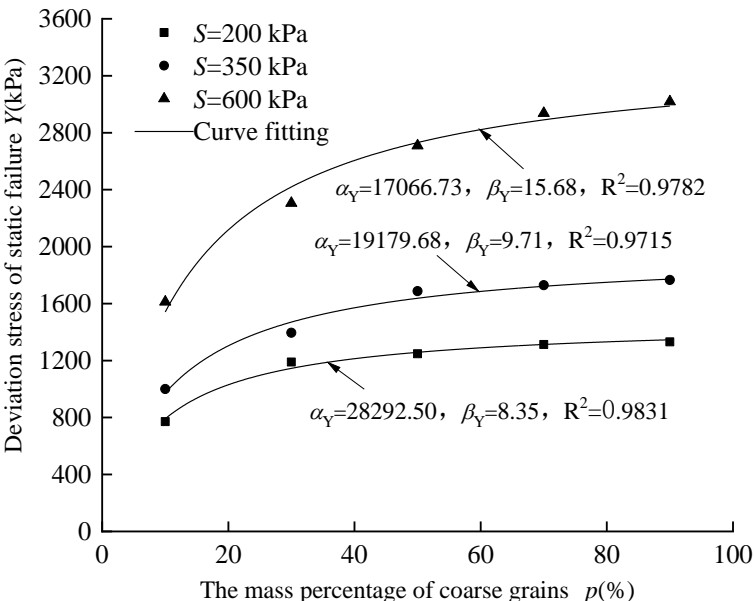

**Figure 9.** Relationship curve between the deviation stress of static failure and the mass percentage of coarse grains.

The following hyperbolic relationship was used to fit the relationship between deviation stress of static failure $Y$ and the mass percentage of coarse grains $p$ under a certain confining pressure, as expressed in Equation (2):

$$Y = \frac{\alpha_Y p}{\beta_Y p + 1} \tag{2}$$

where $\alpha_Y$ and $\beta_Y$ are the fitting coefficients of the fitting relationship of deviation stress of static failure,

The fitting results are shown in Figure 9. For the fitting curves with a confining pressure $S$ of 200, 350 and 600 kPa, the square of the fitting correlation coefficients were 0.9831, 0.9715 and 0.9782, respectively.

### 3.3. The Influence of Coarse Grain-Content on Cohesion and Internal Friction

The shear strength of the soil was determined by the friction and cohesion effect between the grains, which met the Mohr–Coulomb theory; the expression is expressed in Equation (3):

$$\tau = \sigma \tan \varphi + c \tag{3}$$

where $\tau$ is the shear stress on the shear failure surface, that is, the shear strength of soil, $\sigma$ is the normal stress on the shear failure surface, $\varphi$ is the internal friction angle, and $c$ is cohesion.

For the triaxial compression test, $\tau$ and $\sigma$ were calculated as Equations (4) and (5), respectively:

$$\tau = \frac{1}{2} Y \cos \varphi \tag{4}$$

$$\sigma = \frac{1}{2}(Y + 2S) - \frac{1}{2} Y \sin \varphi \tag{5}$$

According to Equations (3)–(5), the Mohr–Coulomb criterion on the deviation stress of static failure can be obtained, which is expressed in Equation (6):

$$Y - (Y + 2S) \sin \varphi - 2c \cos \varphi = 0 \tag{6}$$

For soil samples with different confining pressures, in the $\tau$-$\sigma$ coordinate system, the limit state Mohr circle is drawn according to the deviation stress of static failure $Y$ and the confining pressure $S$, and three confining pressures correspond to three limit state Mohr circles. According to the limit equilibrium condition, the common tangent of the Mohr circle of these limit states is the Mohr–Coulomb shear strength envelope, the intercept between the envelope and the $\tau$-axis is the cohesion, and the angle between the envelope and the $\sigma$-axis is the inner friction angle. The values of cohesion and internal friction angle are shown in Table 3.

**Table 3.** Fitting results of coefficient.

| The Mass Percentage of Coarse Grains $p$ (%) | Cohesion $c$/(kPa) | Angle of Internal Friction $\varphi$ (°) |
|:---:|:---:|:---:|
| 10 | 86.9 | 31.1 |
| 30 | 133.8 | 36.2 |
| 50 | 107.1 | 40.5 |
| 70 | 89.5 | 42.4 |
| 90 | 84.6 | 43.1 |

Figure 10 shows the relationship between cohesion $c$ and the mass percentage of coarse grains $p$. A, B, C, D and E denote the cohesive values corresponding to different mass percentages of coarse grains, respectively. The curve is mainly divided into two part. The cohesion $c$ increased firstly and then decreased with the increase in the mass percentage of coarse grains $p$. When the mass percentage of coarse grains $p$ was 30%, the maximum cohesion was 133.8 kPa at point $B$. In the initial stage of loading, as is shown in the $O$-$A$ segment of the curve, the content of coarse grains was small, and the relationship between the mass percentage of coarse grains and cohesion was linear. In segment $B$-$E$, with the increase in coarse grain content, the cohesion gradually decreased. When the mass percentage of coarse grains $p$ increased from 10% to 90%, the cohesion $c$ first increased from 86.9 kPa to 133.8 kPa in the $A$-$B$ section, and then decreased to 84.6 kPa in the $B$-$E$ segment.

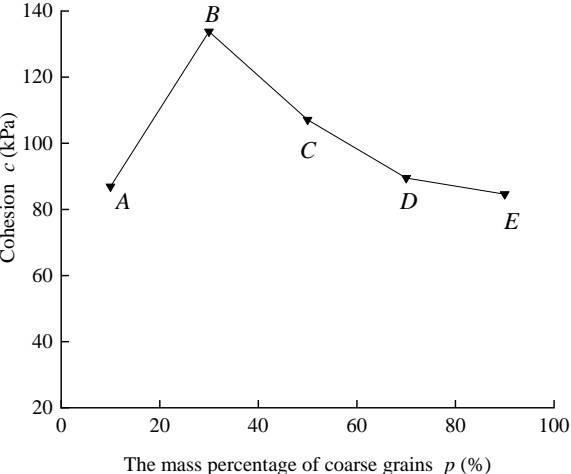

**Figure 10.** Relationship curve between the cohesion and the mass percentage of coarse grains.

Figure 11 shows the fitting results of the relationship between internal friction $\varphi$ and the mass percentage of coarse grains $p$. The following fitting $\varphi$~$p$ relationship was used to define the internal friction angle:

$$\varphi = \frac{\alpha_\varphi p}{\beta_\varphi p + 1} \tag{7}$$

where $\alpha_\varphi$ and $\beta_\varphi$ are the fitting coefficients.

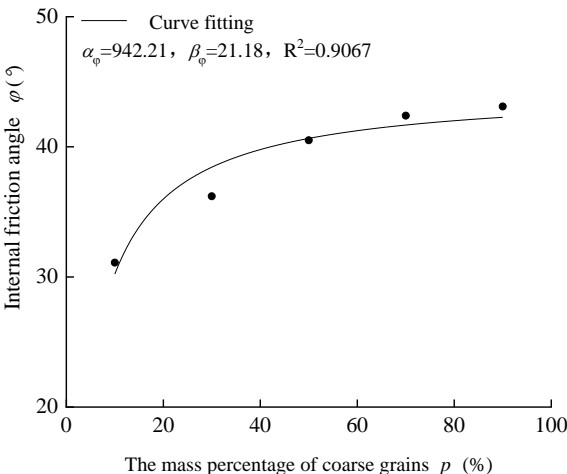

**Figure 11.** Relationship curve between the friction angle and the mass percentage of coarse grains.

The curve is mainly divided into two parts. The cohesion *c* firstly increased and then tended to be stable with the increase in the mass percentage *p* of coarse grains. When the mass percentage of coarse grains *p* increased from 10% to 90%, the internal friction angle $\varphi$ increased by approximately 38.3% from 31.1° to 43.1°. The $\varphi$~*p* fitting results are shown in Figure 11, and the square of the correlation coefficient was 0.9067. In Figures 10 and 11, with the increase in the coarse grain content, the cohesion first increased and then decreased, while the changing trend in the internal friction angle was similar to the deviation stress of static failure.

### 3.4. The Influence of Coarse Grain Content on the Elastic Modulus

The elastic modulus is defined as *E*, which is obtained by axial stress and strain. Figure 12 shows the relationship *E*~*p* between the elastic modulus and mass percentage of coarse grains under different confining pressures. Under different confining pressures, the elastic modulus first increased and then decreased with the increase in the content of coarse grains, and the maximum elastic modulus was at *p* = 30%. When the confining pressure *S* was equal to 200 kPa, the elastic modulus first increased from 863.2 MPa to 1457.9 MPa and then decreased to 1094.5 MPa. When the confining pressure *S* was equal to 350 kPa, the elastic modulus first increased from 1256.8 MPa to 1816.5 MPa and then decreased to 1094.5 MPa. When the confining pressure *S* was equal to 600 kPa, the elastic modulus *E* first increased from 1412.3 MPa to 2058.6 MPa and then decreased to 1735.6 MPa.

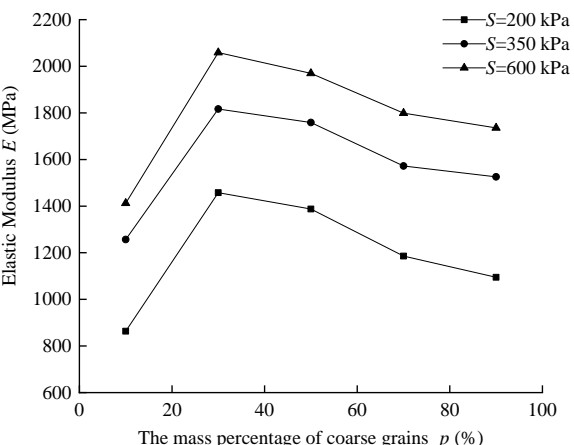

**Figure 12.** Relationship between the modulus and the mass percentage of coarse grains.

### 3.5. The Influence of Coarse-Grain Content on Shear Volume Change

Volumetric strain refers to the ratio of the volume change in the sample to the original volume. The axial strain is defined as $\varepsilon_1$ and the volumetric strain is defined as $\varepsilon_V$. In Figure 13, the typical curves of volumetric stress–strain for the confining pressure of 600 kPa are shown in Figure 13c. The soil grains on the shear failure zone produced obvious dislocation and overturn, so the volumetric strain of the coarse-grained red sandstone soil sample was first induced by compression and then expansion.

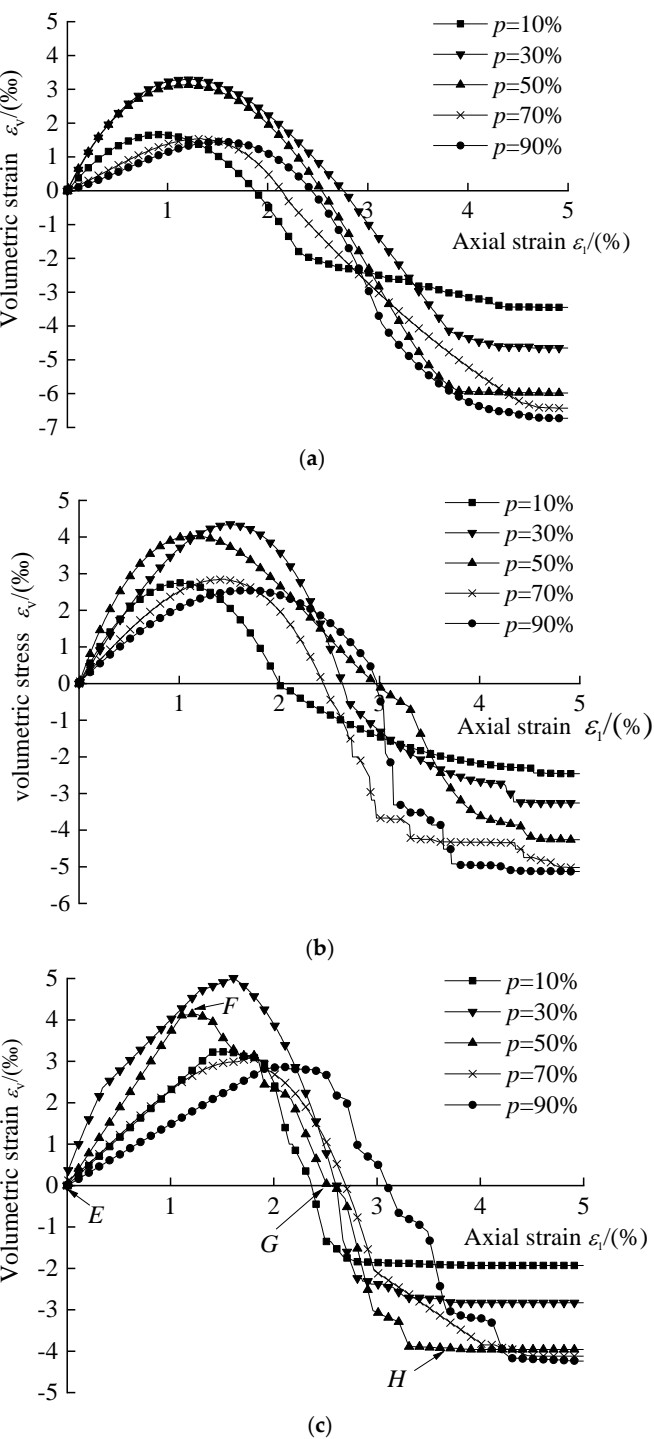

**Figure 13.** Relationship curve between volumetric strain and axial strain (**a**) *S* = 200 kPa, (**b**) *S* = 350 kPa, (**c**) *S* = 600 kPa.

Analyzing the curve corresponding with coarse grain content of 50%, the curve can be divided into three segments. At the initial stage corresponding to segment *E-F*, with the increase in $\varepsilon_1$, the pores of the soil sample were gradually compacted and $\varepsilon_V$ increased. When $\varepsilon_V$ was greater than 0, the sample was at the compression stage. With the further development of shear strain, the volumetric strain reached the maximum point *F*. At the stage corresponding to the *F-H*, the volumetric strain began to change from compression to expansion.

As the volume strain decreased and passed point *G*, the volume strain began to show negative values. At the stage corresponding to the *G-H*, the volume expansion further developed.

The maximum shear shrinkage of the loading process is defined as $\varepsilon_{v\max}^+$. In Figure 14, the relationship between the maximum shear shrinkage $\varepsilon_{v\max}^+$ and the mass percentage of coarse grains *p* under different confining pressures can be seen. As the mass percentage of coarse grains *p* increased from 10% to 90%, $\varepsilon_{v\max}^+$ first increased and then decreased under different confining pressures, and the maximum values were obtained at *p* = 30%. When the confining pressure *S* was 200 kPa, the maximum shear shrinkage increased from 0.00167 to 0.00331 and then decreased to 0.00146. When the confining pressure *S* was 350 kPa, the maximum shear shrinkage increased from 0.00275 to 0.00435 and then decreased to 0.00256. When the confining pressure *S* was 600 kPa, the maximum shear shrinkage increased from 0.00304 to 0.00501 and then decreased to 0.00287. Therefore, for different confining pressures, the change in cohesion was similar. With the increase in coarse grain content, the maximum shear shrinkage first increased and then decreased.

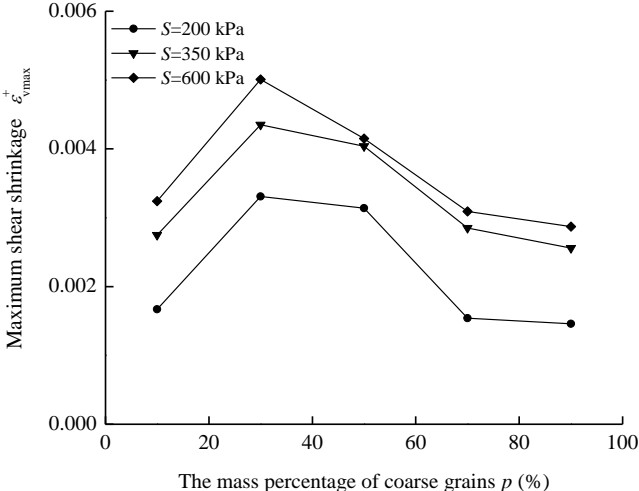

**Figure 14.** Relationship between the maximum shear shrinkage and the mass percentage of coarse grains.

The maximum dilatancy of the loading process is denoted as $\varepsilon_{v\max}^-$ (Absolute value of the maximum volumetric dilatancy strain). In Figure 15, the relationship between the maximum dilatancy $\varepsilon_{v\max}^-$ and the mass percentage of coarse grains under different confining pressures can be seen. Under different confining pressures, when the mass percentage of coarse grain *p* increased from 10% to 90%, $\varepsilon_{v\max}^-$ showed an increasing trend. In addition, for a certain confining pressure, the maximum dilatancy increased with the increase in coarse grain content, but the increase rate gradually decreased. When the confining pressure *S* was 200 kPa, the maximum dilatancy increased monotonically from 0.00345 to 0.00673, with an increase of approximately 95.1%; when the confining pressure *S* was 350 kPa, the maximum dilatancy increased monotonically from 0.00246 to 0.00513, with an increase of approximately 108.5%; when the confining pressure *S* was 600 kPa, the maximum dilatancy increased monotonically from 0.00193 to 0.00424, with an increase of approximately 119.7%. Therefore, similar to the change law of deviation stress of static

failure and internal friction angle, the maximum dilatancy increased approximately in a hyperbolic relationship with the increase in coarse grain content.

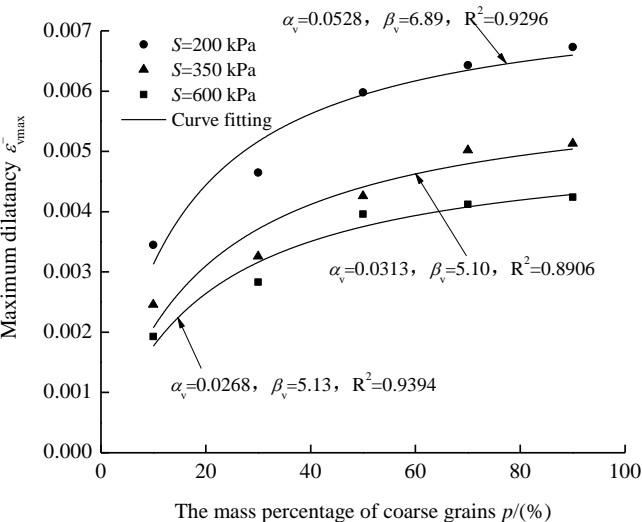

**Figure 15.** Relationship between the maximum dilatancy and the mass percentage of coarse grains.

Considering the approximate hyperbolic relationship between the maximum dilatancy and the mass percentage of coarse grains, the following equation was used to fit the $\varepsilon_{\mathrm{vmax}}^{-}{\sim}p$ relationship under a confining pressure; the fitting results are shown in Figure 15. In Figure 15, the square of the fitting correlation coefficient was 0.8906~0.9394, which indicates that the fitting relationship was good.

$$\varepsilon_{\mathrm{vmax}}^{-} = \frac{\alpha_{\mathrm{v}} p}{\beta_{\mathrm{v}} p + 1} \tag{8}$$

where $\alpha_v$ and $\beta_v$ is the fitting coefficient of the maximum dilatancy fitting equation.

The coarse-grained red sandstone soil samples underwent a two-volume change process, namely shrinkage and dilatancy. Volume changes are often accompanied by slipping, climbing and breakage of the grains. The change law of soil volume and deviation stress of static failure were similar, which are determined by the combination of fine grains and coarse grains. When the ratio of fine grains to coarse grains reaches a certain value, the soil can be compressed to its densest state. For this test, soil with $p$ = 30% could be compressed to the densest state or the largest shear shrinkage. As the variation in soil shear shrinkage is shown in Figure 14, when $p$ was less than 30%, fine grains played a major role, and the shear shrinkage increased with the increase in coarse grain content; when the coarse grain content $p$ was greater than 30%, the coarse grains played a major role, and the dilatancy phenomenon caused by the turn-over of grains became obvious. Therefore, with the increase in coarse grain content, the shear shrinkage of soil decreased. Similarly, in Figure 15, with the increase in coarse grain content, the bond strength between particles was exceeded; therefore, the amount of grain breakage will increase, and the larger grains will be broken into smaller grains to fill the pores, thereby inhibiting volume expansion, and the maximum dilatancy of soil increases and gradually tends to be stable.

## 4. Influence Mechanism Analysis of Coarse Grain Content

In summary, the content of coarse grains had a specific influence on the mechanical parameters of coarse-grained red sandstone soil, such as deviation stress of static failure, cohesion, internal friction angle, maximum shrinkage, maximum dilatancy and elastic modulus. When $p$ increased from 10% to 90%, the deviation stress of static failure, internal friction angle and maximum dilatancy of the coarse-grained red sandstone soil increased, and the growth trend tended to be steep at first and then gently and gradually tended to be

stable. The maximum shear shrinkage, cohesion and elastic modulus of coarse-grained red sandstone soils occurred at $p = 30\%$.

During the test, the pore water of the coarse-grained red sandstone soil sample was drained. The shear strength of soil increases, and the axial stress increases linearly. As the axial strain continues to increase, the soil grains gradually roll, interlock, and overturn, which leads to the dilatancy trend in specimen volume and the axial stress gradually increases to the peak value. When the axial strain increases further, some grains in the soil are broken [23]; at the same time, the axial stress decreases gradually after the shear fracture zone is formed [24], and the volume expansion of the soil is dominant due to the overturning of soil grains. When the strain continues to increase, the stress tends to be stable, and it gradually shows strain-softening characteristics. The volume expansion of the soil also reaches the maximum and tends to be stable, which is shown in Figures 8 and 13, respectively.

The fine particles and coarse grains in the soil will jointly affect the performance of the soil. As the grain content gradually increases, the soil gradually forms a skeleton, the soil performance is gradually dominated by the coarse-grained soil, and the deviation stress of static failure increases. Coarse-grained soils have a greater grain size, irregular shape, poor grinding roundness and a strong interlocking effect between soil grains. This results in an increase in the angle of internal friction. At the same time, the volume expansion of soil caused by the displacement between soil particles becomes gradually obvious. With the further increase in coarse-grained soil, the soil performance is completely controlled by coarse grains. The interlocking effect and friction effects between soil grains become greater, and the deviation stress of static failure increases. However, the fracture phenomenon of the edge of the soil particles is obvious, and the contact of the particles is adjusted. The larger grains are broken into smaller grains to fill the pores [25], which inhibits the increase in the deviation stress of static failure, internal friction angle and dilatancy. The result is that the deviation stress of static failure, internal friction angle and dilatancy of the soil shows hyperbolic growth with the changing trend in coarse grain content, as is shown in Figures 9, 11 and 15, respectively.

The cohesion of coarse-grained soil is related to the effective contact area between the soil grains. The elastic modulus of the soil is mainly determined by the initial arrangement state of the soil grains. The denser the soil, the more closely arranged the soil grains, the greater the effective contact area between the soil grains, and the better the resistance to deformation. That means that the more closely arranged the soil grains are, the stronger the cohesive effect and the greater the elastic modulus. Shear shrinkage is mainly caused by grain breakage, extrusion of small grains in the shear process and cementation breakage between grains [26]. It can be seen from Table 2 that when the mass percentage of coarse grains $P$ was 30 %, the dry density of the soil was the greatest. At this time, the arrangement of soil grains was in the closest state, the effective contact area between soil grains was the largest, the cementation between grains was obvious, the deformation resistance was high, and the soil particles were not easy to break. The cohesion, elastic modulus and shear shrinkage of the soil were also the largest. When the mass percentage of coarse grains $p$ was greater than 30%, the dry density of soil gradually decreased, resulting in a decrease in soil compactness. With the effective contact area between soil grains being reduced, the cementation effect was weakened, the resistance to deformation was reduced, and some soil grains were broken. Soil cohesion, elastic modulus and shear shrinkage were also reduced. Therefore, the cohesion, elastic modulus, and shear shrinkage of the soil with the changing trend in coarse grain content demonstrated the law of first increasing and then decreasing, as is shown in Figures 10, 12 and 14, respectively.

According to the Mohr–Coulomb criterion, the shear strength of soil is mainly determined by the cohesion and friction effect of soil, which are reflected by the cohesive force and internal friction angle, respectively. According to Guo [20], when the mass percentage of coarse grain is less than 30%, fine grains surrounding the coarse grains play a major role in determining the shear strength, and both the cohesive effect and friction effect are influ-

encing the shear strength. When the mass percentage of coarse grains exceeds 30%, coarse grains play a major role in determining the shear strength, and only the friction effect is dominant. Therefore, with the increase in coarse grain content, the cohesion first increased and then decreased, and the changing trend in internal friction angle was consistent with the deviation stress of static failure, and the laws of these two mechanical parameters are shown in Figures 10 and 11, respectively.

## 5. Discussion

As a result of the large number of coarse grains in red sandstone soils, it is necessary to discuss the influence of coarse grains on the mechanical properties of red sandstone soils. Therefore, this paper systematically analyzed the influence of coarse grain content on the strength properties, as well as the deformation characteristics, of red sandstone soils and quantitatively described the relationship between the content of coarse grains in red sandstone coarse-grained soils and these macro-mechanical parameters.

According to Xu et al. [27] and Chen [28], the cohesion and internal friction angle of coarse-grained soils are positively related to the content of coarse grains. Figure 11 shows that the cohesive force of red sandstone coarse-grained soil increased rapidly and then slowly, with the increase in coarse grain content, which is generally consistent with the literature. Figure 10 shows that the cohesive force of the red sandstone coarse-grained soil increased and then decreased with the increase in coarse-grain content, which is different from the results in the literature. The dry density achieved its maximum value when the mass percentage of coarse grains ($p$) was 30%. At this point, the soil was denser and the effective contact area between the soil particles was at its maximum, showing that the cohesive force first increased and then decreased. According to Peng et al. [29], the compaction process of red sandstone coarse-grained soils was divided into a compacting stage, a breaking stage, and a stabilizing stage. As is shown in Figure 13, the volume strain of the sample first demonstrated shear shrinkage, then dilatation, and finally stabilized during the compression process, which corresponds to the three stages outlined in the literature.

The results of this paper indicate that a coarse grain content of 30% is the key value influencing the mechanical properties of red sandstone soils. This can provide some theoretical guidance for practical filling projects. However, this paper is limited to discussing the relationship between the coarse grain content and the mechanical properties of red sandstone soils through macro-mechanical tests and lacks the study of micro-phenomena within the samples. It is important to further study the relationship between the coarse grain content and the mechanical properties of red sandstone soils by combining macro-mechanical tests with micro-mechanical tests.

## 6. Conclusions

As coarse-grained red sandstone soil is widely distributed in the southwest region of China, the influence of the content of coarse grains on the mechanical properties of coarse-grained red sandstone soil was explored based on the consolidated drained triaxial test. The mechanical properties of the coarse-grained red sandstone were determined through several triaxial tests with different percentages of coarse grains under three levels of confining pressures.

During the test, the coarse-grained red sandstone soils showed strain-softening characteristics. The volume first tended towards shear shrinkage, and as the axial strain increased, the soil grains moved, overturned, and broke, and the volume strain changed from shear shrinkage to dilatancy. In addition, the maximum volume shear shrinkage at the same confining pressure was reached at 30% coarse grains.

It was found that 30% coarse grains was the key value for the mechanical properties of the coarse-grained red sandstone soil. When the mass percentage of coarse grains was 30%, the cohesion reached 133.8 kPa with an internal friction angle of 39 degrees, and the elastic modulus of the coarse-grained red sandstone soil was 1816.5 MPa, corresponding to a confining stress of 350 kPa. The coarse grains improved the shear strength of the



red sandstone soil if the mass percentage was increased to 30%; thus, a value of 30% is recommended for embankment construction when coarse grains are used to improve the mechanical properties of red sandstone soil.

**Author Contributions:** Investigation, J.C. and Y.Z.; resources, J.C.; data curation, J.C. and X.J.; writing—original draft preparation, Y.Z. and Y.Y.; writing—review and editing, J.C., Y.Y. and B.Y.; visualization, B.Y. and B.H.; methodology, Y.Y. and B.Y. All authors have read and agreed to the published version of the manuscript.

**Funding:** This work was supported by the Research Basic Ability Improvement Project of Young and Middle-aged Teachers in Guangxi of China (No. 2020ky05036). This work was supported by the Foundation for Science and Technology Base and Talents of Guangxi Provence of China (GUIKE AD21220051). This work was supported by the Innovation Project of GUET Graduate Education of China (No. 2021YCXS180).

**Institutional Review Board Statement:** Not applicable.

**Informed Consent Statement:** Not applicable.

**Data Availability Statement:** Not applicable.

**Conflicts of Interest:** The authors declare no conflict of interest.

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
