# Peer review of "Influence of Coarse Grain Content on the Mechanical Properties of Red Sandstone Soil"

_sustainability, doi:10.3390/su15043117_

Round 1

Reviewer 1 Report

1- The works in the introduction were only mentioned. Expand them with more details. For example:

“Wang et al.[12]analyzed the dilatancy characteristics of coarse-grained soil through large-scale triaxial test”.

“Babenko [13] and Voznesensky et al. [10] studied the stress-strain and volumetric compression characteristics of coarse-grained soils through large-scale triaxial tests”.

Expand them and write about the results of their works.

2-Rewrite this statement:

“In this paper, the influence of coarse-grained content on the mechanical properties of coarse-grained red sandstone soil is considered for experimental purpose, and a large triaxial instrument is used to carry out the triaxial shear test”.

3- Rewrite this statement:

“Through the experiments, the influence of coarse-grain content on mechanical parameters such as deviation stress of static failure, cohesion, internal friction angle, elastic modulus, dilatancy, and shrinkage of coarse-grained red sandstone soil was analyzed, and the influence mechanism was discussed, which provided a reference for the study of mechanical properties and engineering applications of coarse-grained red sandstone soil”.

4-Mention the gap of previous studies and the main goal of this study in the last paragraph of the introduction section.

5-Why the soil compaction ratio of 0.9 was selected? For example, why not 0.95?

6- In Table 1, change “he” to “the”.

7- What were the standards for soil gradation and triaxial tests?

8- How did the values of Table 1 obtained? Explain sample preparation, standard, test, …

9- The English should be improved through manuscript. For example this sentence is wrong: “Define the confining pressure as S”.

10-What kind of triaxial test was used? It was CD?

11- Why the loading rate is 0.3% per minute? How did you determine it?

12-Based on Table 1, the mass percentage of coarse grain (p) of 50% had the highest Maximum dry density. Why in the triaxial tests (Fig.8), the mass percentage of coarse grain of 90% showed the highest strength?

13-What were the relative density (Dr) and void ratio (e) of Red Sandstone Soil?

14- Mention the properties of soils.  They were taken from where? Which city or area? What were their physical and chemical properties? What were their specific gravity (Gs)? …  

15- Explain Table 2. Why this trend for cohesion and friction happened? Why the p=30% had the highest cohesion and p=90% had the highest friction?

16- How did you calculate the maximum volumetric dilatancy strain? Show the process of calculation for one of them for example.

17- Compare the results of this study with the results of previous works in the discussion section.

18-The number of references are few. Use more and new references.

Author Response

Dear Reviewer:

Thank you for the reviewers' comments concerning our manuscript entitled "Influence of Coarse-grain Content on Mechanical Properties of Red Sandstone Soil'' (ID: sustainability-2111757). Those comments are all valuable and very helpful for revising and improving our paper, as well as the important guiding significance to our research. We have studied the comments carefully and have made corrections which we hope meet with the reviewers' approval.

We have provided a point-by-point response to the reviewer’s comments and revised the manuscript using the "Track Changes" function. Please see the attachment.

We appreciate for Reviewers' warm work earnestly and hope that the correction will meet with approval.

Once again, thank you very much for your comments and suggestions.

Sincerely yours,

All authors

Reviewer 2 Report

Dear Authors,

I have some following comments and questions:

·         What is the reason for cohesion initially increasing due to the increase of the content of coarse grains of red sandstone?

·         Please provide some photographs of the soil used in the research.

·         How was the soil compaction ratio determined? It is crucial in analyses presented in the manuscript. Please provide the table or graph of compaction ratio all the 15 samples used in the research. The information that the compaction ratio equals 0,9 is insufficient.

·         Word mistake in Table 1: “he mass percentage of coarse grain”

·         In lines 339-342 the Authors stated that some grains in the soil are broken. Was that phenomenon investigated? If not, it should be explained based on the literature or previous research and referred.

·         In lines 357-360 the references as necessary to draw that conclusion.

·         In line 366 the Author stated that shear shrinkage is mainly caused by grain breakage. So how can explain that shrinkage is observed at small stresses? References needed.

·         I disagree with the conclusion in lines 399-400 especially with: “the coarse-grained red sandstone soil became contractive”. In my opinion, it should be less dilative. I do not see the contractive behaviour of investigated samples. Initially, volume decreasing is common in both dilative and contractive behaviour.

Author Response

(The authors gave the same response as above.)

Round 2

Reviewer 1 Report

The English should be improved. For example, lines 430-433 and 440-442 should be rewritten.

Author Response

Dear Reviewer:

Thank you for the reviewers' comments concerning our manuscript entitled "Influence of Coarse-grain Content on Mechanical Properties of Red Sandstone Soil'' (ID: sustainability-2111757). We appreciate your recognition of the content of our manuscript research.

We have chosen MDPI's language editing service to enhance English. The language of the manuscript has been enhanced. We have revised the manuscript using the "Track Changes" function. Please see the attachment.

We appreciate for Reviewers' warm work earnestly and hope that the correction will meet with approval.

Once again, thank you very much for your comments and suggestions.

Sincerely yours,

All authors

Reviewer 2 Report

Dear Authors,
Thank you for clarifying your work and including all the comments in your manuscript. In this form, this manuscript is very comprehensive and pleasant to read and I hope that it will be for other scientists. 
Best regards

Author Response

Dear Reviewer:

Thank you for the reviewers' comments concerning our manuscript entitled "Influence of Coarse-grain Content on Mechanical Properties of Red Sandstone Soil'' (ID: sustainability-2111757). We appreciate your recognition of the content of our manuscript research.

We are grateful to the reviewers for their previous comments and suggestions.

Sincerely yours,

All authors